# Integrating Physical Activity into a Nutrition and Exercise Science Middle School Curriculum: The THINK Program

**DOI:** 10.3390/nu17091538

**Published:** 2025-04-30

**Authors:** Arlette Perry, Joseph Bonner, Sophia Williams, Wei Xiong, Alejandro Garcia, Carolina Velasquez, Alexis Friedman, Debbiesiu L. Lee, Ingrid de Lima Hernandes, Ji Shen, Marisol Meyer, Lucia Fernandez

**Affiliations:** 1Department of Kinesiology and Sport Sciences, University of Miami, 5202 SW University Drive, Coral Gables, FL 33146, USA; jbonner1@umiami.edu (J.B.); sar379@miami.edu (S.W.); ingrid.lima.hernandes@gmail.com (I.d.L.H.); 2Department of Educational and Psychological Studies, University of Miami, 5202 SW University Drive, Coral Gables, FL 33146, USAdebbiesiu@miami.edu (D.L.L.); mxm2560@miami.edu (M.M.); lxf455@miami.edu (L.F.); 3Department of Experimental Psychology, University of Sao Paulo, Sao Paulo 05508-220, Brazil; 4Department of Teaching and Learning, University of Miami, 5202 SW University Drive, Coral Gables, FL 33146, USA; j.shen@miami.edu

**Keywords:** nutrition science, exercise science, physical fitness, body composition, physical activity, integrative health

## Abstract

Background/Objectives: Recent interest has emerged in novel initiatives that focus on the “whole child” to improve the health and well-being in youth. The purpose of this study was to determine whether a translational health in nutrition and kinesiology (THINK) program addressing physical, psychological, and educational well-being could improve personal health and lifestyle behaviors in youth. Methods: A total of 81 adolescents (44 males, 37 females, 12.50 + 0.62 years) were evaluated at the beginning and end of the spring semester across three different years: 2019, 2021, and 2023. The physical literacy measures included the Physical Activity Enjoyment Scale (PAES) and the Physical Activity Perception Scale (PAPS), along with knowledge-based tests in nutrition and exercise science. Social emotional learning (SEL), STEM education, and nutrition habits were evaluated using standard surveys and questionnaires. The physical evaluations included measures of body composition and physical fitness. Results: At the completion of the spring semester of each year, evaluations were compiled for all three years, with decreases found for % body fat (*p* < 0.001) and increases found for lean body mass (*p* < 0.001). The physical fitness components, including muscular strength (*p* < 0.001), cardiorespiratory fitness (*p* < 0.001), power (*p* < 0.001), and flexibility (*p* < 0.01), all improved. The physical literacy components, including the PACES (*p* < 0.001), PAPS (*p* < 0.001), exercise (*p* < 0.05), and nutrition science knowledge (*p* < 0.05) results, also improved. There were no significant changes in any other variables. Conclusions: A novel in-school academic curriculum integrating the physical, psychological, and educational well-being of the whole child could lead to improved body composition, physical fitness, and physical literacy.

## 1. Introduction

During the last 10 years, educational goals have focused on the “whole child”, emphasizing knowledge (cognitive gains), social (affective components), and physical (psychomotor development) factors aimed at promoting healthy lifestyle behaviors [1]. Despite the push for “whole-child” initiatives in the United States (U.S.), the current obesity rate in American youth is at an all-time high of 20.6% [2], with <25% of youth meeting the national guidelines for physical activity participation [3]. While 225 min of physical education is recommended each week, only 7.9% of middle school students meet these requirements [4]. The downstream effects of poor nutrition and inadequate exercise are adverse medical conditions, including pre-diabetes, hypertension, asthma, and fatty liver disease, which continue to plague youth in the U.S. [5,6,7]. To reduce these concerns, other programs have attempted to integrate nutrition and physical activity into the classroom environment. For example, Planet Health [8] targeted middle school adolescents, integrating health-related units into traditional classes (e.g., language arts, math, science, and social studies), concomitant with brief micro-sessions in physical education. Among girls and boys, it was effective for reducing sedentary behaviors and increasing curriculum-based knowledge. In girls specifically, it was more effective in reducing obesity, improving weight-control behaviors, and improving healthy food options [9]. Another program integrating health and nutrition into an existing science curriculum, along with physical education or aerobic exercise classes, was effective in reducing body fat, insulin resistance, and inflammatory markers in 8th grade adolescents [10]. The Sports, Play, and Active Recreation for Kids (SPARK) program [11] is a student-oriented physical education program with an expanded sport and fitness component focused on physical activity enjoyment. The program features a variety of sports, managerial skills, and take-home newsletters to foster more parental involvement. After a 2-year period, the cardiorespiratory fitness improved in both traditional and standard physical education programs. Physical competence and exercise enjoyment, which are components of physical literacy, improved in the SPARK program only.

Several limitations have been found in each of these programs. For example, in Planet Health [8,9], neither STEM nor SEL was integrated into the curriculum and physical fitness was not evaluated. In the 2nd program, STEM was integrated into the program and health-related markers were examined and improved. However, physical fitness was not evaluated, and physical activities were not integrated into the academic lessons but rather offered independently [10]. The cardiorespiratory fitness did improve over time in both traditional physical education and SPARK [11] classes; however, there was no clear difference in results between the two programs. The physical activity participation was greater in the SPARK [11] group. It did appear that younger more than older adolescents in the SPARK program performed better on their cardiorespiratory fitness scores and in their physical competence over time. Unfortunately, there were no academic lessons in nutrition or exercise science, no evaluation regarding the benefits of active lifestyle behaviors, and no hands-on laboratory or clinical evaluations in either program.

The Translational Health in Nutrition and Kinesiology (THINK) summer program was a sport and fitness curriculum that included nutrition and exercise lectures, SEL, and STEM. The three components were integrated into one program designed to facilitate positive health behaviors in middle school adolescents [12]. Improvements were found in physical fitness, social emotional learning (SEL), and nutrition habits. However, this was not an academic program and no improvements in STEM attitudes were observed. Furthermore, there was no testing or grading; therefore, it was unknown whether the participants successfully learned concepts in nutrition and exercise science or the importance of adopting healthy lifestyle behaviors. The in-school THINK program is an academic curriculum born out of the need for a more holistic strategy for improved physical, psychological, and educational well-being. Since many schools in the United States are facing tremendous pressure to improve math, reading, and standardized test scores, health and physical education requirements have been reduced, thereby limiting the nutrition and physical activity options [13]. The proposed THINK program is an integrative curriculum that embraces the whole-child approach. The program includes an academic component focusing on nutrition and exercise science, a laboratory component that applies nutrition and exercise science information, and a physical activity component that reinforces active lifestyle behaviors. It was hoped this program would enable students to “think” about their nutrition and exercise choices, empowering them to improve their own personal health behaviors. The program includes a social, affective component enhancing SEL that promotes positive health behaviors, along with an educational component that embraces STEM education in an applied manner. Elements of physical literacy [14], referred to as the confidence, competence, and promotion of active lifestyle behaviors, along with a knowledge of the benefits of an active healthy lifestyle, were included as part of the curriculum. This program was built on the translation of research in nutrition and exercise science into a classroom setting that also integrates physical activities into an academic curriculum.

The purpose of this study was to examine a spring semester THINK program covering a 3-year period: 2019, 2021, and 2023. During this time, a program that included nutrition and exercise science, SEL, and STEM education were integrated into an academic curriculum that also included physical activity. The program was implemented as an independent, stand-alone course taken for a grade, which served as part of the 7th grade curriculum.

## 2. Materials and Methods

Having recently been extended to include one 7th and 8th grade class, the principal of one elementary school wanted to initiate a community–university partnership. The intervention middle school possessed the standard required academic classes (math, English, science, history, etc.); however, similar to many middle schools in the United States, physical and health education were not required in the 7th grade. Since the school was geographically close to a major university that could provide expertise in teaching and learning and the principal had heard about the THINK project from earlier pilot programs [15,16], the investigators were approached about implementing an innovative curriculum for this school. Following several meetings, the partnership moved forward for implementation. The exclusion criteria included failure to complete assent or consent forms and failure to attend classes in person or being physically unable to participate in the physical activities. Across the 3-year period, 100% of the students were eligible to participate in this study. Following school approval, approximately 90 students were available to complete the program. However, several students moved, a few sustained orthopedic injuries preventing them from participating, and a few students were dismissed for disciplinary reasons, leaving a total of 81 students available for testing. Not all students completed the pre- and post-testing, as a few were absent, ill, or required to complete standardized exams during the time of the program’s testing. This study was conducted in accordance with the Declaration of Helsinki, and approval for the study was provided by the University of Miami’s Institutional Review Board for the Use of Human Subjects (protocol #20170693), the Miami Dade County Public School Board (protocol #2303), and the school principal.

All participants were required to complete baseline and post-testing measurements to be included in the data analysis. Attendance was assessed daily, and the final grades reflected weighted scores on school quizzes, laboratory assignments, and physical activity participation. The THINK program started in January and ran for five months (one semester), ending in May of the same year. Data were collected and analyzed for the years of 2019, 2021, and 2023. Two years were omitted because in 2020 no outside programs were allowed in any public schools due to COVID, and during 2022 the Dade County Public School approval occurred after the semester had already began. During the approved years, classes ran two days/week from 13:00 to 15:00 for a total of four hours/week. The baseline and follow-up testing ran for 1–2 weeks at the beginning and end of the spring semester.

Faculty members from their respective areas of expertise in Kinesiology, SEL, and STEM education worked with graduate assistants to review pilot THINK projects that had been published elsewhere [17,18]. The lead investigators and their assistants prepared an orientation and training workshop to acquaint the research assistants with the goals of the program, the outcome variables and corresponding testing protocols, and the testing procedures to ensure the reliability and consistency of the data collection. Along with the faculty, the research team was introduced to each educational theme and asked to develop different physical and laboratory activities that reinforced the educational information presented. The participants were required to provide both baseline and post-testing data that included demographic information, anthropometric measurements, physical fitness variables, and self-reported surveys and questionnaires.

### 2.1. Physical Measurements

Height was recorded using a wall-mounted stadiometer while weight was measured on an electronic scale from the Inbody 570 machine (InBody, Co. Ltd., Seoul, Republic of Korea). The body mass index (BMI) was calculated using the formula weight (kg)/height (m)^2^. The InBody 570 features multi-frequency segmental bioelectrical impedance to measure the total adiposity and muscle mass. In adolescent populations, this machine has been validated against underwater weighing with fair correlations reported (r = 0.79 for girls and (r = 0.69) for boys (*p* < 0.10 for both) [19].

### 2.2. Physical Fitness Measures

Physical fitness included both health and skill-related fitness components encompassing muscular strength, cardiorespiratory endurance, muscular endurance, power, and flexibility. The validity and reliability of all physical fitness variables have been reported elsewhere (12). Muscular strength was evaluated using a digital hand-held dynamometer (Camry Scale, El Monte, CA, USA). While in a seated position with the dominant hand bent at a 90-degree angle, the participants squeezed the dynamometer as hard as possible using the dominant arm [20].

Cardiorespiratory fitness was evaluated via the 20-Meter Fitnessgram Pacer Test, wherein students ran as many laps as possible within a specified time allotted for each lap until the student can no longer keep pace completing laps [21].

Muscular endurance was evaluated by measuring the number of “curl-ups” the student could complete according to the procedures outlined by the President’s Council on Physical Fitness and Sports [22]. During testing, participants were asked to perform as many curl-ups as possible within one minute starting from a supine position, with their feet and gluteus region separated by 12 in.

Lower-body power was tested using the Vertec TM Jump Training System (Jump USA, Sunnyvale, CA, USA). Each adolescent’s standing height with their arm stretched overhead was subtracted from their maximal jump height. The jump entailed one step forward and a squat countermovement performed to the desired depth prior to jumping [23]. Flexibility was tested by evaluating the range of motion of the lower back and hamstring muscles using a Sit-and-Reach Box (Acuflex I, Novel Products Inc., Rocktown, IL, USA) [24]. While the subjects were seated on the floor with their outstretched legs abutting the box, they were asked to slowly move a lever as far forward as possible without bending the knees.

Agility was evaluated using the 30-Foot Eraser Shuttle-Run, in a protocol recommended by the President’s Council on Physical Fitness and Sports [22]. The participants were required to run as quickly as possible to retrieve two small board erasers from the opposite end, picking up one eraser at a time, while covering a 30-foot distance. The time to completion was recorded only after the second eraser was successfully placed at the starting point.

### 2.3. Self-Reported Questionnaires/Surveys

Nutrition Habits: The Adolescent Food Habits Checklist (AFHC) survey [25] was used to address specific dietary practices with which adolescents tend to have more control of eating behaviors. These included the consumption of calorically dense foods, low-fat alternatives, and fruits and vegetables, as well as snacking behaviors. The 23-question checklist provided answers of “true”, “false”, or “doesn’t apply to me”. The AFHC is reported to be a reliable and valid self-reported survey for assessing dietary habits in adolescents. A previous investigation of 1882 adolescents showed scores on this scale to possess a Cronbach’s α = 0.82 and a high test–retest reliability of r = 0.91 [25].

SEL: This was evaluated using the Positive Youth Development Inventory [26], a 58-item questionnaire that assesses five subscales of youth development: character, competence, confidence, connection, and contribution. The scores ranged from a minimum of one to a maximum of four points per question. Investigators have reported a reliability coefficient of Cronbach’s α = 0.92 [26].

STEM Attitude: The Attitude towards STEM Questionnaire, 6–12th Grade Edition [27], was used to determine specific levels of interest within various components of STEM subject areas, encompassing education and careers. A psychometric analysis of 2500 middle school students found an internal consistency and reliability value of Cronbach’s α = 0.89 for the total survey, with values of 0.75 for the science component, 0.89 for the mathematics component, and 0.77 for both the engineering and technology components. The test–retest reliability value was Cronbach’s α = 0.80 for the total survey, 0.70 for science, 0.89 for mathematics, and 0.77 for engineering and technology [28].

Physical Literacy: The Physical Activity Enjoyment Scale [29] was used to evaluate motivation and how positively respondents felt about the physical activities they were engaged in. The subjects provided responses to being physically active (I enjoy it…My body feels good…It gives me energy…I have fun…Using a rating scale from…I strongly agree…I strongly disagree). The scale has a maximum score of 100 points and a reported internal consistency of 0.93 using Cronbach’s alpha [29]. To measure exercise confidence and competence, participants were asked to rate how well they felt they would perform during each of the physical fitness tests [30]. All physical fitness measures were carefully described to the students prior to allowing them to rate how they would perform. A Likert scale was used to evaluate corresponding ratings: (1) poor; (2) below average; (3) average; (4) above average; (5) excellent. The perceived competence in each physical fitness skill was summed and the average score of all six physical fitness components was computed. To measure exercise science knowledge, a 25-item exam was given on similar groups of topics covering physical fitness, warm-ups and stretching, the heart, thermoregulation, and metabolic fitness. For example, in one question, the students were asked to complete the three main risk factors associated with coronary heart disease. In another question, the students were asked to list at least two health-related and two skill-related variables.

To measure nutrition science knowledge, a 25-item exam encompassing topics such as micronutrients and macronutrients, label reading, and hydration was given. For example, in one question, the students were asked how much fluid one should replenish with if they lost 4 lbs of body weight after exercising outdoors in the heat and humidity. Other questions involved calculations of the total and saturated fat contents in a serving of a particular food. Although nutrition knowledge is not technically considered part of the definition of physical literacy, a good portion of the nutrition information addressed sports nutrition. Therefore, we included this measure as part of the physical literacy assessment. Both the nutrition and exercise science tests were developed by the teacher and director of the program to cover key information taught in the classroom. Although the tests were presented as raw scores for the data analysis, both tests were curved for grading purposes.

### 2.4. Program

The THINK program is based on three integrated components: (1) educational sessions encompassing health-related themes; (2) laboratory experiences rooted in STEM education; (3) structured physical activities that promote sportsmanship, camaraderie, and fun. All three components were integrated into each session. Therefore, health-related themes encompassing topics on physical fitness, the heart, metabolic fitness, nutrition, human anatomy, and physiology were introduced initially. This was followed by hands-on laboratory experiences that reinforced the thematic lectures. Finally, physical activities were designed to reinforce both the health-related themes and laboratory experiences. For example, during the warm-up and stretching lecture, information was discussed regarding the types, timing, and benefits of warm-ups and stretching. During the laboratory portion, students used goniometers to measure the ranges of motion of the upper and lower body to see how the ranges of motion can differ or be site-specific across bodily segments. Finally, during the physical activity portion, student teams designed warm-ups for different sports (e.g., basketball, soccer) that were performed by the entire class after the warm-up. The integration of thematic lectures and laboratory and physical activities into each class is one unique feature of the THINK curriculum. Furthermore, all health-related themes began with probing questions that prompted the students to reflect upon their responses, promote student engagement, and encourage student interaction. Specific, measurable, attainable, realistic, and timely (SMART) goals were always discussed at the conclusion of each lecture to facilitate individual approaches for improving personal health and lifestyle behaviors. These included overcoming barriers to exercise, attaining realistic goals, and empowering students to improve their physical and mental health. The aforementioned elements were woven into the lectures, laboratory experiences, and physical activities, promoting a unique integration of physical, psychological, and educational well-being in a single course. We know of no other programs integrating all three elements into one academic program. During the exercise science lectures, the director of the program along with doctoral students in exercise physiology taught the lectures. To address STEM-related information, a STEM faculty member from the Teaching and Learning Department taught the students how to analyze data generated in exercise science. To promote positive behavioral changes, a faculty member and her doctoral students from the Educational and Psychological Studies Department discussed reducing barriers to exercise, goal setting, and other methods to promote beneficial lifestyle changes.

For example, the student goals included eating more fruits and vegetables, eating less “junk” food, and exercising more. The self-reported perceived barriers to reaching their goals included not being in charge of buying food for themselves and having little time for exercise in the midst of academic pressures and family expectations. When pinpointing barriers, the program facilitators brainstormed ways in which the students could advocate for their own health through conversations with parents and caregivers, and setting goals that were reasonable and manageable within given timeframes. The laboratory and clinical experiences enabled the students to learn how to use goniometers to measure their range of motion; calipers to measure their skinfold thickness; dynamometers to measure their muscular strength; Douglas flow meters to measure their respiratory function; and field tests to measure their agility, aerobic fitness, and balance. STEM education was incorporated into several laboratories wherein the students generated scatterplots for several physical variables they participated in and asked if physical skills such as muscle strength and endurance, flexibility, and power were or were not related. Several laboratories featured a competitive element such as “Simon says”, wherein the students had to point to the correct bone (ulna, femur, or acromion process) or muscle (deltoid, gastrocnemius, or triceps) to stay in the game. The laboratory experiences always reinforced the health-related themes.

Nutrition education was an important element of the curriculum. The classroom presentations focused on relevant nutrition topics such as processed versus unprocessed foods, nutrient-dense foods, deciphering macronutrients from micronutrients, alternative plant-based options, hydration, and reading labels. During this part of the lecture, food models were used to enhance the nutrition education. Certain food items were introduced into the classroom, such as a 20 oz bottle of coke, wherein a volunteer added 13 teaspoons of sugar to allow the students to visualize the carbohydrate content of this drink. Educational resources were introduced into the classroom using pads and posters, along with models, to improve the students’ awareness of portion sizes. Throughout the units, the investigators initiated practical conversations about ways to make school lunches healthier. During the heart and metabolic health lectures, the class was introduced to the Mediterranean meal plan and the importance of a well-rounded diet featuring a variety of vegetables and fruits.

Physical activities were incorporated into the program, featuring outdoor games and activities that reinforced health-related themes (see Table 1). For example, during the nutrition lecture, the students created and then compared the MyPlate poster to the Harvard Healthy Eating Plate poster. During the physical activity portion, the participants had to sprint to a shopping bag containing groceries, randomly pick an item from the shopping bag, then sprint to a MyPlate poster and place the selected grocery store item under the appropriate food group. Points were awarded to the fastest teams with the most accurate placement of food items.

The SEL component was drawn from the social cognitive theory [31] and grounded in the general conceptual model, reinforcing the fact that positive relationships between physical activities and physical fitness can be translated to one’s quality of life and health [32,33]. The methods for promoting and sustaining positive behavioral changes included identifying emotions, making responsible choices, setting up reasonable goals, communicating with others, reaching out for help, and empowering youth to make positive nutrition and exercise choices. Based on the social cognitive theory [32], our belief was that if we could facilitate the students’ self-efficacy (their beliefs in their capacity to achieve healthy behaviors), we could promote their actions in this area. In each of our targeted competencies (physical and nutritional health, SEL, STEM), we sought to enhance their self-efficacy by imparting the students with critical knowledge, modeling desired behaviors, and demonstrating and providing them with opportunities to practice the desired skills. The curriculum provided a nurturing, socially positive environment that encouraged youth to “think” about their growth and actions in all aspects of their life. This included instilling confidence to make healthy decisions regarding nutrition and exercise options, developing competence in academics, and promoting communication and self-reflection while building positive relationships.

### 2.5. Statistical Analyses

All collected data were analyzed using the SPSS statistical package (version 27, IBM SPSS Inc., Armonk, NY, USA). The mean values and standard errors of the means (SEMs) were determined for all physical and physical fitness variables, as well as completed questionnaires and surveys. The normality of the distribution for physical and physical fitness variables was determined via visual and numerical assessments, with the criterion for skewness and kurtosis being + 2.00. A paired samples *t*-test was conducted at baseline and at the program’s completion to measure changes in variables at the end of the THINK program.

## 3. Results

### 3.1. Subject Characteristics

Table 2 shows the subject characteristics. The average age of the participants was 12.5 years. A total of 53% of the students were Latino/x and 78.3% were Latino/x and Black. There were more males in the study, with 54.2% of attendees being male and 45.8% of attendees being female.

### 3.2. Anthropometric and Body Composition Characteristics

Presented in Table 3 are the body composition measurements. The BMI values were average with no significant changes observed following the program. Percent body fat however, dropped 8% from 22.76 ± 1.11 to 20.87 ± 1.06 (*p* < 0.001), at the program’s completion. Furthermore, lean body mass rose 5.5% from 90.19 ± 2.09 lbs to 95.23 ± 1.86 lbs (*p* < 0.001) at the end of the program.

### 3.3. Physical Fitness Measures

Shown in Table 4 are the physical fitness levels presented for six variables. The grip strength increased 7.3% from 23.78 ± 2.0.98 kg to 25.52 ± 1.08 kg, (*p* < 0.001). Scores on the PACER test increased 24% from 25.32 ± 1.68 laps to 32.13 ± 1.83 laps, (*p* < 0.001). Lower body power increased 20.9% from 13.7 6 ± 0.35 to 16.64 ± 0.40, (*p* < 0.001). The flexibility increased 6.7% from 19.13 ± 1.21 cm to 20.42 ± 1.26 cm, (*p* = 0.002). There were no improvements in any other physical fitness variables.

### 3.4. Self-Reported Survyes and Questionnaires

Shown in Table 5 are the results of the self-reported questionnaires. The exercise enjoyment increased 4.9% from 3.26 ± 0.09 to 3.42 ± 0.09, *p* = 0.011. The exercise perception (competence) increased 11.7% from 3.40 ± 0.08 to 3.80 ± 0.10, *p* < 0.001. The nutrition knowledge raw scores rose 33.2% from 51.07 ± 5.93 to 68.03 ± 7.98, *p* = 0.019. The exercise science knowledge improved 24.8% from 54.48 ± 7.55 to 73.05 ± 3.29, *p* = 0.019. There were no improvements in any other self-reported surveys or questionnaires.

## 4. Discussion

Following the THINK program, significant improvements were found in body composition. The students experienced an 8% decline in body fat, which was encouraging, since at 12.5 years, the adolescents were likely to be in the middle of puberty, wherein gains in adiposity are most dramatic [34]. Globally, obesity is a major public health concern, with one out of every 10 individuals aged 5–17 years being either overweight or obese [35]. A meta-analysis of obesity prevention programs in youth showed that obesity negatively and disproportionately impacts health among minority ethnicities, especially in high-income countries such as the United States [36]. The inclusion of indices other than BMI that more directly measure adiposity and its components has been recommended [37]. The results of the meta-analysis appeared to be more positive when a combination of personalized physical activity, more culturally appropriate healthy food options, and more portion guidance was included in the programs. Finally, programs that engage families, the community, and stakeholders comprising policy makers and health professionals appeared to work best [36,37]. Although the THINK program was primarily designed to be a health promotion not necessarily an obesity prevention program, it featured interactive and structured play activities and healthy, traditional food options, along with portion size awareness. Furthermore, community stakeholders including school administrators and health professionals, who were directly engaged in the program, were included. These elements may have contributed to the significant reductions in adiposity in our program, since they are considered successful strategies for reversing obesity and its associated co-morbidities [36]. In addition to reductions in adiposity, the students experienced a 5.5% increase in lean body mass, which has also been associated with greater health benefits. In a meta-analysis of youth, increases in lean body mass marked by muscular fitness were associated with greater bone health and decreases in cardiovascular risk factors, insulin resistance, and inflammatory markers [38]. Participation in outdoor sport activities demanding high energy expenditure, such as obstacle course relays and competitive team sports, may have contributed to the beneficial findings in terms of body composition.

Physical fitness improvements complemented the changes in body composition. Following the program, increases in strength, aerobic fitness, lower-body power, and flexibility were found. Significant improvements in strength and power are indices of neuromuscular function that are shown to track well into adulthood. Furthermore, neuromuscular benefits are associated with improvements in skeletal mass, bone density, and cognitive processing [39], all of which serve to enhance functional independence later in life [39,40]. The positive gains in neuromuscular measures were likely due to the creative relays, unique obstacle courses, and circuit training, all of which were included in this study. The cardiorespiratory fitness increased by 24%. Research has shown that even small improvements in cardiorespiratory fitness translate into large reductions in cardiovascular risk later in adulthood [41]. The research shows that after controlling for gender, maturational age, and body composition, Hispanic and African-American youth are found to possess lower peak VO2 values than non-Hispanic White youth [42]. Since nearly 80% of participants in our study were Hispanic and Black, the improvements in cardiorespiratory fitness were particularly compelling. These positive results were likely related to the outdoor aerobic activities and numerous team sports such as soccer, basketball, and flag football that were integrated into the classroom. Increases in range of motion have also been associated with health benefits, including reduced arterial stiffness, greater vascular compliance, and improved endothelial function [43]. Therefore, increases in lower back and hamstring flexibility may have positive implications for cardiovascular health in the future. All outdoor physical activities began with a group warm-up and ended with group stretching, which may have benefitted the flexibility outcomes.

Overall, the participants improved in four out of six physical fitness variables. With only one-third of the total class time allotted to physical activity, positive fitness gains were observed. This may have been related to the interdisciplinary nature of the curriculum and the integration of physical activities into a program that also includes hands-on, clinical experiences and lectures related to personal health and active lifestyle behaviors. High physical fitness levels in youth are considered a powerful predictor of cardiometabolic health [44] and reduced rates of premature cardiovascular disease [45].

The stakeholders, particularly the school principal and assistant principal, worked closely with the college faculty to support the program. The school reserved 2-h block scheduling for the implementation of the THINK curriculum. This extra time facilitated the integration of physical activities into the academic curriculum and was very helpful for promoting positive gains in physical fitness.

One of the unique aspects of this program was the interdisciplinary nature of this curriculum. The nutrition and exercise science information was reinforced by hands-on laboratory experiences designed to apply the information learned in the classroom. The sport and exercise activities emphasized the translation of both lecture and hands-on experiences to active lifestyle behaviors. The SEL curriculum was integrated into the program by introducing modules centered around adopting a growth mindset, reducing barriers to exercise, establishing healthy lifestyle goals, and achieving a positive interface within the community. Unfortunately, no significant improvements were found in SEL. The research suggests that some domains of self-concept tend to be stable and resistant to change, whereas others are more likely to change within middle schoolers [46,47]. Further, previous research suggests that there may be significant individual variations in how self-concept changes during this developmental period [48]. In the current study, SEL was measured using the Positive Youth Development Inventory, which captures both stable self-concepts (i.e., “I am a creative person”) and those that are more amenable to change (i.e., “I can manage my emotions.”). Therefore, this measure may not have been sensitive enough to capture the specific changes made in SEL. In future research, investigators plan to utilize an SEL measure that more closely aligns with the competencies targeted within the SEL curriculum.

In contrast to the in-school program, the THINK summer program resulted in significant improvements in SEL. However, the summer program was 7 h/day, 4 days/week for six weeks. The extra time allotted for SEL learning was used to divide the classes into smaller groups for more personal attention. Perhaps concepts of SEL need to be presented in smaller group formats for the delivery of SEL information. In the present study, the SMART goals enhanced student engagement; however, the time was limited for student interaction. This may have reduced the positive reinforcement of SEL activities and impacted the results.

Despite the targeted effort and significant increases in both nutrition and exercise science knowledge and learning, there was no observed change in the students’ attitudes toward STEM. The measurement of STEM attitudes, however, reflects only one aspect of STEM learning. Furthermore, limited STEM time was provided in the THINK program compared with STEM classes students have taken throughout the years. This may have contributed to the lack of significant findings. There are many factors that can influence the students’ attitudes toward and interest in STEM [49,50], and many of these factors are hard to change in a short period of time. The STEM lectures were also adapted from college material, which may have been too ambitious for these students. Compared to the outdoor exercises and activities that the students favored and enjoyed, the STEM topics may have been too challenging. Therefore, additional time may be needed to absorb STEM materials at this age. More research is needed to further understand how to nurture student attitudes toward STEM in an interdisciplinary learning environment.

Although the students’ nutrition knowledge increased by 33%, their nutrition habits did not significantly change. Given the different stages of readiness for change, family influences, and the environment, it has been reported to take anywhere from 66 days to 24 months to initiate positive changes in eating habits [51]. Other studies targeting the lunchroom cafeteria (canteen) showed evidence of the complexity and tremendous difficulties in improving healthy nutritional options at this age [52,53]. During the THINK summer program, their nutrition habits did improve; however, there was more time devoted to practical nutritional activities such as taste testing and on-site lunches, so as to reinforce healthy food options. The concentrated time devoted to discussing healthy nutrition options and nutrition-related activities during the summer may have been helpful in eliciting significant changes.

Interestingly, significant gains were found in key components of physical literacy such as exercise enjoyment, perception, and knowledge regarding the benefits of exercise. The participants improved in all four components of physical literacy, accounting for some of the strongest improvements in the study. Concomitant with large gains in nutrition and exercise science knowledge, positive changes in physical literacy can facilitate greater physical activity participation and the promotion of healthy lifestyle behaviors in adolescents. The overall results remained remarkably consistent in terms of outcome variables during each semester of study. The Physical Activity Enjoyment Scale (29) did not show significant improvements in 2021 and the physical fitness perception (30) showed a trend toward significance in 2019. The gains in physical fitness were fairly consistent, with the exception of flexibility, which did not significantly change in 2021. The major difference in physical fitness variables was in the strengths of significance, which varied from *p* < 0.05 to *p* < 0.001 across the three years of study. The measures of SEL and STEM attitudes were also very consistent, not showing any significant changes during each of the three years of study.

It should be noted that the average age of subjects was 12.5 years, which represents a critical period for the rapid growth, quantity, and distribution of fat and muscle during adolescence [34]. On average, girls start puberty earlier than boys and show higher adiposity and lower muscle mass levels than boys [54]. We did not use Tanner staging [55] to classify pubertal development in our study, nor did we find gender differences in any physical fitness variables. Our findings were similar to other studies showing mixed results regarding gender depending upon the level of skill and the consistency, type, and length of intervention [56,57]. Our study was also limited to one semester; therefore, the magnitude of physical maturation may not have been great enough to impact gender differences in outcome measures.

### Limitations

It is necessary to acknowledge several limitations within this study. These included the small sample size of 81 students in one school, which limited the external validity. Although Latino/x students comprised the majority of the students in the study, as well as in Miami Dade County Public Schools [58], these results may not be generalizable to all schools in the South Florida community. A control group of adolescents attending a more traditional 7th grade class should be examined to determine whether these significant changes were due to the program itself or to innate biological growth and physical maturation. In the absence of a control group, the pre–post changes in Tanner staging could be accounted for statistically to determine whether puberty contributed to the significant changes observed during the program. It may, however, be more feasible to simply add a control group of adolescents and compare changes in outcome variables between the THINK and control groups. The guidelines for success in reducing adolescent obesity include parental involvement [37]. Other than sending assignments home, whereby parents had to assist in securing food labels to their child’s favorite food and help to identify joint parent–child physical activities, there was limited parental involvement in the THINK program. As recommended by previous investigators [36,37], it would be important to examine several related cardiometabolic outcomes such as basal insulin, glycosylated hemoglobin, lipid or lipoprotein, and blood pressure levels.

Improvements in food habits, SEL, and STEM attitudes were not observed. This may have been related to using questionnaires and surveys that did not fully align with the targeted competencies that the program addressed. Perhaps a greater understanding of the factors that uniquely motivate students to make healthy choices is warranted. Therefore, it may work best to have students identify the factors that uniquely motivate them toward certain actions and incorporate those factors into the intervention. The self-determination theory [59] may be a useful framework for identifying how students’ intrinsic and extrinsic factors motivate their healthy choices. Recent studies have used innovative methods to identify common motivating factors that drive behavior [60]. Furthermore, a better distribution of the time allocated to SEL and STEM may be necessary. The STEM information modified from college nutrition and exercise science classes may have been too challenging for the students. Simplifying the topics or reducing the number of topics presented may have enabled the students to better process the information discussed in class.

An analysis of variance between the groups across time did not show gender differences in any outcome variables. Although sufficient power was achieved to determine gender differences in physical fitness variables, the inclusion of additional subjects and a more equivalent number of females and males may be necessary to better examine gender differences during adolescence. Finally, there was no follow-up of the program. In a previous pilot of the THINK summer program, positive gains in physical fitness were maintained 6 months after completion of the THINK summer program [16]. Plans to include a longitudinal follow-up using more schools to extend our findings would enhance the comprehensiveness and quality of the program

## 5. Conclusions

Our research adds to the existing literature supporting the “whole-child” approach to education, integrating physical, psychological, and educational well-being into a single academic curriculum. Although the combination of lecture and hands-on activities reduced the total time devoted to physical activity participation, improvements in both body composition and physical fitness were observed. Given the global concerns regarding obesity that are prevalent in minority populations, particularly those from high-income communities, the combination of decreased adiposity concomitant with increased lean body mass during adolescence is particularly meaningful. Improvements in both aerobic and neuromuscular fitness are essential to long-term cardiovascular benefits and functional independence. Gains observed in physical literacy are integral to engaging in regular physical activity throughout the lifespan. Our study represents an important step forward translating the “whole-child“ approach to the classroom setting to facilitate positive lifestyle behaviors early on in youth.

## Figures and Tables

**Table 1 nutrients-17-01538-t001:** Physical Activities in Relationship to Health Related-Topics.

Timeframe	Activities
Weeks 1–2: Physical, Health, Skill Related Fitness	Physical & Health Related Fitness - Tug of War - Yoga - Soccer - Basketball - Kickball - Dodgeball Physical & Skill Related Fitness - Circuit Training - Relay Races - Agility Drills
Weeks 3–4: Warm-up & Stretching	Dynamic & Sports-specific Warmups - Soccer warm-up - Basketball warm-up - Gymnastics warm-up - Volleyball - Bosu ball balance activities
Weeks 5–6: Nutrition	Nutrition - Nutrition education relay races - Nutrition questions during individual activities: relay races, individual sports, skill-related fitness activities
Weeks 7–8: Heart Health	Heart Health - Jogging - Basketball - Soccer - Circuit training: burpees, jumping jacks, jogging in place, high knees, lunges Students took their own manual heart rates
Weeks 9–10: Metabolic Fitness	Metabolic Fitness - Potato sack races - Hula-Hoops - Calisthenics - Light weight circuits
Weeks 11–12: Muscle & Bones	Muscle & Bones - Jump-rope skipping - Plyometric box jumping - Hurdle jumping - Light medicine ball throws

**Table 2 nutrients-17-01538-t002:** Subject Characteristics at Baseline (*n* = 81).

Subject Characteristics	Baseline
Age (years)	12.54 ± 02
Height (cm)	160.02 ± 0.89
Weight (kg)	55.03 ± 1.65
**Race/Ethnicity (%)**	
Black	25.02
Hispanic	53.26
White	21.72
**Gender (%)**	
Male	54.17
Female	45.83

Note: Mean values for age, weight, and weight are listed. Percentages by race (out of 100%) and gender (out of 100%) are also categorized for the entire sample.

**Table 3 nutrients-17-01538-t003:** Anthropometric and Body Composition Characteristics at Baseline and Post-testing.

Characteristics	Baseline	Post-Testing	*p*-Value
BMI (kg/m^2^) (*n* = 77)	21.39 ± 0.62	21.18 ± 0.59	0.153
BMI z score (*n* = 77)	−0.003 ± 0.16	0.023 ± 0.2	0.754
BMI Percentile (%) ≥85th	38.32%	38.03%	0.364
Body Fat (%) (*n* = 77)	22.76 ± 1.11	20.87 ± 1.06	<0.001
Lean Body Mass (kg) (*n* = 77)	90.19 ± 2.09	95.23 ± 1.86	<0.001

Note: Data are presented as means ± standard error of the mean. *n* = 81 unless otherwise indicated. BMI, body mass index.

**Table 4 nutrients-17-01538-t004:** Physical Fitness Measures at Baseline and Post-testing.

Characteristics Physical Fitness	Baseline	Post-Testing	*p*-Value
Strength (kg) (*n* = 81)	23.78 ± 0.98	25.52 ± 1.08	<0.001
Aerobic Fitness (laps) (*n* = 75)	25.32 ± 1.68	32.13 ± 1.83	<0.001
Power (cm) (*n* = 76)	13.76 ± 0.35	16.64 ± 0.40	<0.001
Flexibility (cm) (*n* = 77)	19.13 ± 1.21	20.42 ± 1.26	0.002
Agility (s) (*n* = 75)	10.92 ± 0.20	10.79 ± 0.19	0.137
Curl-ups (#) (*n* = 78)	45.17 ± 1.58	45.45 ± 1.80	0.864

Note: Data are presented as means ± standard error of the mean. Strength reported using grip dynamometry. Aerobic fitness assessed using the Pacer Test and number of laps completed. Power reported as maximum jump height. Flexibility recorded using a sit and reach test. Agility assessed using Shuttle Run Test. Curl-ups evaluated the number performed in one minute (raw score).

**Table 5 nutrients-17-01538-t005:** Self-Reported Surveys/Questionnaires and Exams.

Self-Reported Surveys/ Questionnaires	Baseline	Post-Testing	*p*-Value
SEL (*n* = 64)	3.16 ± 0.03	3.17 ± 0.04	0.735
STEM Attitude (*n* = 47)	3.68 ± 0.09	3.64 ± 0.11	0.680
Nutrition Food Habits (*n* = 75)	13.80 ± 0.61	14.49 ± 0.58	0.190
Exercise Enjoyment (*n* = 78)	3.26 ± 0.09	3.42 ± 0.09	0.011
Exercise Perception (*n* = 78)	3.40 ± 0.08	3.80 ± 0.10	<0.001
Nutrition Knowledge (*n* = 63)	51.07 ± 5.93	68.03 ± 7.98	0.019
Exercise Knowledge (*n* = 63)	54.48 ± 7.55	73.05 ± 3.29	0.019

Note: Data are presented as means ± standard error of the mean. *n* = 81 unless otherwise indicated. SEL, Social Emotional Learning, STEM, Science, Technology, Engineering, Mathematics. Social emotional learning was assessed via the Positive Youth Development Inventory on a 1–4 pt Likert rating scale. STEM, Science, Technology, Engineering & Math was assessed via The Students Attitudes Towards STEM Survey (Version 6–12th Grade) on a 1–5 pt Likert rating scale with the average total score listed. Nutrition habits were scored via the Adolescent Food habits Questionnaire based on 1 pt for each healthy response, out of a maximum of 23 questions. Physical literacy was assessed via the Physical Activity Enjoyment Scale (Exercise Enjoyment), Fitness Ability Perception Scale (Exercise Perception), a nutrition knowledge-based examination, and an exercise knowledge-based examination. Exercise Enjoyment and Fitness Ability Perception were scored on a 1–5 Likert rating scale with 1 being the lowest and 5 being the highest. The nutrition and exercise knowledge exams were scored on a 0–100% scale.

## Data Availability

The data presented in this study are available from the corresponding author on request. The data are not publicly available due to ethical reasons.

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
