# Peer review of "Integrating Physical Activity into a Nutrition and Exercise Science Middle School Curriculum: The THINK Program"

_nutrients, 2025, doi:10.3390/nu17091538_

Round 1

Reviewer 1 Report

Comments and Suggestions for Authors

I would like to thank for the opportunity to review this manuscript. Please see the following comments to consider to further increase the quality of this manuscript.

The absence of a control group limits the ability to attribute improvements solely to the THINK program. Future iterations should include a comparison group of students receiving a standard curriculum to control for natural developmental changes.

Authors have mentioned several times term motivation, but have not provided any relevant theoretical background. For example, the self-determination theory is one leading theory of motivation. The advantage of the self-determination theory is that it differs from different forms of motivation such as intrinsic motivation and extrinsic motivation. Future research would do well by relying on a specific motivational behaviors list to develop an intervention program to attempt to actually change health behaviors. For example, in a recent high level study by Ahmadi et al., (2023), a classification system of motivational behaviors was developed. This list of motivational behaviors was developed based on Delphi method, gaining input from the very best of motivational research all around the world. I believe that combining methods from AI and knowledge from motivational studies could significantly change health behavior among schoolchildren. For example, in a recent high level research by Ravšelj et al., (2025) it was found that students feel positive emotions, with curiosity and calmness being the most common reactions. Building on this research by Ravšelj et al., (2025), I think that AI-tailored approach to health behaviors such as individual-based motivational strategies with the aim to increase intrinsic motivation towards health behaviors could be highly recommended.

Ahmadi, A., Noetel, M., Parker, P., Ryan, R. M., Ntoumanis, N., Reeve, J., Beauchamp, M., Dicke, T., Yeung, A., Ahmadi, M., Bartholomew, K., Chiu, T. K. F., Curran, T., Erturan, G., Flunger, B., Frederick, C., Froiland, J. M., González-Cutre, D., Haerens, L., . . . Lonsdale, C. (2023). A classification system for teachers’ motivational behaviors recommended in self-determination theory interventions. Journal of Educational Psychology, 115(8), 1158–1176. https://doi.org/10.1037/edu0000783

Ravšelj D, Keržič D, Tomaževič N, Umek L, Brezovar N, A. Iahad N, et al. (2025). Higher education students’ perceptions of ChatGPT: A global study of early reactions. PLoS ONE 20(2): e0315011. https://doi.org/10.1371/journal.pone.0315011

The study was conducted in a single school with a relatively small sample size (n=81), limiting external validity. Expand the program to multiple schools with diverse demographics to improve the generalizability of results.

While physical fitness and body composition improved, the manuscript glosses over why SEL and STEM outcomes did not show significant changes. Provide a more robust discussion on potential reasons for these null findings. For example, consider whether the time allocated to SEL and STEM activities was insufficient or whether the measurement tools were sensitive enough to detect changes.

The manuscript acknowledges the lack of long-term follow-up but does not propose strategies to address this. Include a follow-up phase in future studies to assess the sustainability of physical and behavioral improvements beyond the immediate post-intervention period.

While most results are well-presented, some areas, such as the differentiation between years (2019, 2021, 2023), lack clarity. Clearly indicate if results varied across the three years or if they were aggregated, and justify the approach.

The average participant age (12.5 years) falls within a critical period for pubertal development, which could confound results. Acknowledge and discuss the role of biological maturation as a confounding factor in body composition and fitness improvements.

Author Response

Comments 1: The absence of a control group limits the ability to attribute improvements solely to the THINK program. Future iterations should include a comparison group of students receiving a standard curriculum to control for natural developmental changes.

Response 1: We really appreciate this reviewer’s comments and feel this was the biggest limitation to the study. We included plans for a comparison group under conclusions pg. 14 lines 606-611.

Comments 2: Authors have mentioned several times term motivation, but have not provided any relevant theoretical background. For example, the self-determination theory is one leading theory of motivation. The advantage of the self-determination theory is that it differs from different forms of motivation such as intrinsic motivation and extrinsic motivation. Future research would do well by relying on a specific motivational behaviors list to develop an intervention program to attempt to actually change health behaviors. For example, in a recent high level study by Ahmadi et al., (2023), a classification system of motivational behaviors was developed. This list of motivational behaviors was developed based on Delphi method, gaining input from the very best of motivational research all around the world. I believe that combining methods from AI and knowledge from motivational studies could significantly change health behavior among schoolchildren. For example, in a recent high level research by Ravšelj et al., (2025) it was found that students feel positive emotions, with curiosity and calmness being the most common reactions. Building on this research by Ravšelj et al., (2025), I think that AI-tailored approach to health behaviors such as individual-based motivational strategies with the aim to increase intrinsic motivation towards health behaviors could be highly recommended.

Response 2: We really appreciated the reviewers input here and I spent a great deal of time discussing this with our EPS expert and co-author. We also appreciate your keen attention to detail and our use of the term “motivation” in several places throughout the manuscript. We are very excited about the possibility of incorporating self-determination theory as a guiding framework for our future work.  In our current research, our guiding theoretical framework was social cognitive theory. Our underlying belief was that, if we could facilitate students’ self-efficacy (their beliefs in their capacity for healthy behaviors), we could promote their actions in this area. We added more information about this under our SEL explanation on pg 7, lines 330-335. We also discussed limitations to our use of the Positive Youth Development Scale on pg 12 lines 494-504. Under conclusions, we addressed how we could incorporate the self-determination theory into our own research citing two main references that were suggested on pg 14 lines 595-603.  Finally, we removed all terminology related to “motivation”, and replaced with terminology that more closely aligns with our conceptual framework, so as not to confuse the readership with our guiding theoretical framework. 

Comment 3: The study was conducted in a single school with a relatively small sample size (n=81), limiting external validity. Expand the program to multiple schools with diverse demographics to improve the generalizability of results.

Response 3: We agree wholeheartedly that our small sample size limited external validity of the study and added that to our limitations section on pg 13 lines 558- 559.

Comment 4: While physical fitness and body composition improved, the manuscript glosses over why SEL and STEM outcomes did not show significant changes. Provide a more robust discussion on potential reasons for these null findings. For example, consider whether the time allocated to SEL and STEM activities was insufficient or whether the measurement tools were sensitive enough to detect changes.

Response 4: Again, we agree with the reviewer on this as well as the other reviewers citing the same comment.  Since we did get significant improvements in SEL during our THINK Summer program, we believe, our lack of significant results, was due, in part, to the distribution of time devoted to SEL during the in-school program and also, a limitation in the evaluation tool itself.   Therefore, we discussed potential reasons for SEL results  on pg 12 lines 494-511.

With respect to STEM, we believe lack of results were due mainly to using a tool that evaluates STEM Attitudes more so than STEM learning. Therefore, we clarified that throughout the document.  We also revised the document to state that STEM attitude is only one aspect of STEM learning. Since we never achieved significant improvements using our STEM questionnaire during summer or in school program, we also felt that the level of difficulty may have been too complex for participants and/or simply that more time is needed to absorb the information presented. This information was presented in the discussion section on pg 13 lines 531-543 and again under limitations on pg 14 lines 573-580.

Comment 5: The manuscript acknowledges the lack of long-term follow-up but does not propose strategies to address this. Include a follow-up phase in future studies to assess the sustainability of physical and behavioral improvements beyond the immediate post-intervention period.

Response 5: We agree with the reviewer on this and have submitted a grant that enables us to follow up with multiple schools using multiple classes with control and experimental groups. Both groups of students will be followed up after a 6-month period. We did not want to project beyond that time  since we wish to see what the results are first. This is presented in the conclusions section, pg 14 line 606-613. We also used your suggested terminology of … “expanding the program to include multiple schools with diverse populations to enhance the generalizability of results.” We really wanted to “thank you” for your input on this.

Comment 6: While most results are well-presented, some areas, such as the differentiation between years (2019, 2021, 2023), lack clarity. Clearly indicate if results varied across the three years or if they were aggregated, and justify the approach.

Response 6: Since we could only do one class per year, there was really no specific approach other than to try to get a larger sample size with results that may be more generalizable to the population. However, we also did data analysis on outcome variables each year in the process. The biggest differences found were in the strength of significance for the physical fitness variables along with variables related to physical literacy.  We pointed this out on pg 13 lines 547-555.

Comment 7: The average participant age (12.5 years) falls within a critical period for pubertal development, which could confound results. Acknowledge and discuss the role of biological maturation as a confounding factor in body composition and fitness improvements.

Response 7: Yes! This was the most emphatic request by reviewers. Therefore, we spent a great deal of time examining this further and reporting on other studies examining this in relation to gender differences. We discussed this on pg 11 lines 443-477. We also discussed this in our limitations section on pg 13 lines 564-572.

Reviewer 2 Report

Comments and Suggestions for Authors

 The Manuscript entitled "Integrating Physical Activity into a Nutrition and Exercise Science Middle School Curriculum: The THINK program" presents a study evaluating the effectiveness of the THINK program, an academic curriculum integrating physical, psychological, and educational health for adolescents.

  1. This study lacks a control group of students. How can we determine whether the observed progress is brought about by the project itself or is due to the students' own physical growth and development?
  2. No significant improvements were observed in STEM learning, nutrition, and SEL. What could be the possible reasons?
  3. How is the sample size determined in the research, especially in the analysis of gender differences? A larger sample size will enhance the statistical power and reliability of the research results. Please provide some references.
  4. The abbreviations in the table should be accompanied by their full names below the tables.

Author Response

Comment 1: This study lacks a control group of students. How can we determine whether the observed progress is brought about by the project itself or is due to the students' own physical growth and development?

Response 1: We wish to thank this reviewer for insightful comments and excellent points raised throughout the manuscript. We agree that this is probably the greatest limitation to our study. Unfortunately, it’s hard to find a school willing to volunteer as a control.  We discussed the only  possible way to do this without a control group  under limitations on pg 13 lines 564-572 and again under conclusions on pg 14 lines 606-613. Since we have submitted a grant to follow-up with multiple schools including both experimental and control groups, we feel this is the best way to determine if results are specific to the program itself or to biological growth and maturation. 

Comment 2: No significant improvements were observed in STEM learning, nutrition, and SEL. What could be the possible reasons?

Response 2: We agree that this is another excellent point also raised by other reviewers. The information regarding SEL was presented in the discussion section on pg 12 lines 494-511 and again under limitations on pg 14 lines 573-576. It was also addressed again under conclusions on pg 14 lines 596-603. For STEM, it was addressed on pg 13 lines 531-543 and again under limitations on pg 14 lines 576—580. It was also addressed again on pg 14 lines 603-613. Information regarding nutrition limitations were addressed on pp 12-13 lines 512- 530.

Comment 3: How is the sample size determined in the research, especially in the analysis of gender differences? A larger sample size will enhance the statistical power and reliability of the research results. Please provide some references.

Response 3: This is an important comment. We never did a power analysis to determine differences by gender in any outcome variables. Our goal was to see if we could improve outcome variables in all middle school participants independent of gender. Given our population of middle school adolescents, physical fitness was certainly  one key area wherein we would expect to see gender differences. However, this was not the case in our study. Upon examination of the literature, it appeared to be a little more complicated than we expected. This issue of gender was addressed under discussion on pg 11 lines 443-477 and again, on pg 14 lines 581-585.

Comment 4: The abbreviations in the table should be accompanied by their full names below the tables.

Response 4: Thank you for catching this. This information was taking care of on the tables.

Reviewer 3 Report

Comments and Suggestions for Authors

Strengths:

First of all, this is one of the best interventions that I have read and it fits the Special Issue theme quite well. The purpose, methods and findings certainly have merit. The THINK program and collaborative between Kinesiology and Sport Sciences, Educational and Psychological Studies, and Teaching and Learning, is quite impressive. This is a good program.

Weaknesses / Suggestions:

I give credit to the authors for pointing out the major limitation - that is no real comparison as could have been seen with a control. As an applied piece of work, this is an excellent project. However, as a piece of science, this limitations keeps it from my labeling it as HIGH. 

Other:

1- the abstract and introduction are a bit confusing regarding the design, which is later clarified. I was asking, "At the end of  ‘each’ semester for 3 years across time? Or, it appears, a different 7th grade class tested for each semester over one year, the another class the next year? For 3 year?

2- Do not identify the principal as she / her.. just principal.

3- Did you compare results for sex? compare male and female? For many, this is an important question.

Author Response

Comment 1: The abstract and introduction are a bit confusing regarding the design, which is later clarified. I was asking, "At the end of  ‘each’ semester for 3 years across time? Or, it appears, a different 7th grade class tested for each semester over one year, the another class the next year? For 3 year?

Response 1: We wish to thank this reviewer for their thoughts and encouragement while also pointing out some important inconsistencies. We agree the Abstract and Introduction on our wording of the years we studied. This was also pointed out by another reviewer. We went back to page 1 of the Abstract to clarify this in two places lines16-17 and also lines 23-24. We also clarified it again in the Introduction on pg 2 lines 88-91. A rationale was given for why these years studied on pg 3 lines118-120.

Comment 2: Do not identify the principal as she / her.. just principal.

Response 2: We appreciate this comment. We went back to change this as appropriate on pg 3 lines 94-98.

Comment 3: Did you compare results for sex? compare male and female? For many, this is an important question.

Response 3: We appreciate this comment as every reviewer wanted to know more about gender differences. This was addressed first on pg 11 lines 443-455, again under limitations on pg 13 lines 564-572 and on pg 14 lines 581-585.

Round 2

Reviewer 1 Report

Comments and Suggestions for Authors

Authors have done well job on revising their manuscript.

Author Response

We thank you for providing wonderful feedback on this manuscript. These suggestions certainly improved the quality of this research and submission. 

Reviewer 2 Report

Comments and Suggestions for Authors

The authors have addressed my concerns.

Author Response

(The authors gave the same response as above.)
